# SELF-SPECIALIZATION: UNCOVERING LATENT EXPERTISE WITHIN LARGE LANGUAGE MODELS

## ABSTRACT

Recent works have demonstrated the effectiveness of self-alignment in which a large language model is, by itself, aligned to follow general instructions through the automatic generation of instructional data using a handful of human-written seeds. Instead of general alignment, in this work, we focus on self-alignment for expert domain specialization (e.g., biomedicine, finance), discovering it to be very effective for improving zero-shot and few-shot performance in target domains of interest. As a preliminary, we first present the benchmark results of existing aligned models within a specialized domain, which reveals the marginal effect that "generic" instruction-following training has on downstream expert domains' performance. To remedy this, we explore **self-specialization** that leverages domain-specific unlabelled data and a few labeled seeds for the self-alignment process. When augmented with retrieval to reduce hallucination and enhance concurrency of the alignment, self-specialization offers an effective (and efficient) way of "carving out" an expert model out of a "generalist", pre-trained LLM where different domains of expertise are originally combined in a form of "superposition". Our experimental results on both biomedical and financial domains show that our self-specialized model outperforms its base model by a large margin. Notably, our self-specialized one based on MPT-30B for biomedicine even surpasses larger popular models based on LLaMA-65B, highlighting its potential and practicality for specialization, especially considering its efficiency in terms of data and parameters. Our code will be released upon acceptance.

## 1 INTRODUCTION

Instruction-tuning (Ouyang et al., 2022; Wei et al., 2022; Mishra et al., 2022; Su et al., 2022) of large language models (LLMs) offers a mechanism to adeptly guide models using specific directives, thereby enhancing their versatility across diverse tasks. However, as promising as this concept might seem, it poses an inherent challenge: the substantial need for quality data (Chung et al., 2022; Wan et al., 2023; Köpf et al., 2023). The very premise of instruction-tuning hinges on the availability of well-crafted, human-annotated data, a resource that is both time-consuming and challenging to scale efficiently (Honovich et al., 2022; Kang et al., 2023).

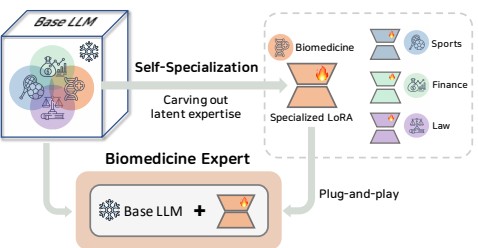

Figure 1: Self-specialization concept. Expertise in various domains is mixed and latent within base LLMs. Target domain expertise is carved out through self-specialization.

Furthermore, acquiring domain-specific data is even more demanding as it requires the involvement of domain experts, which is often more expensive (Bai et al., 2021; Wang et al., 2023).

Emerging as a promising solution to this data-intensive challenge is the approach of self-alignment (Wang et al., 2022a; Sun et al., 2023). By allowing LLMs to automatically generate instructional data from a handful of human-authored seeds, self-alignment presents a means to harness the internal general knowledge of these models (which results from extensive pre-training on the internet corpora (Devlin et al., 2019; Raffel et al., 2020; Brown et al., 2020)) without extensive human annotations.

However, some pertinent questions remain: (i) How effective are the original or the self-aligned models when applied to more niche domains, such as biomedicine? (ii) Given that neither the

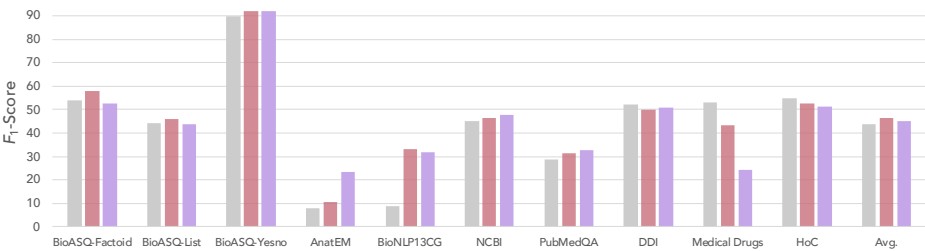

Figure 2: Benchmarking results of a base LLaMA-65B and its aligned variants, Alpaca-65B and Dromedary-65B, on a biomedical domain across 10 datasets, covering various NLP tasks such as question answering, information extraction, classification, etc. 5-shot results are presented.

initial pre-training nor subsequent self-alignment tuning is domain-specific, we hypothesize that the model expertise in different domains resides in "superposition" in the model's parameters and hidden states. In other words, parametric knowledge in LLMs represents a mixture of semantics and knowledge of various domains. May this hinder each individual expert-domain performance? These inquiries become even more practical when considering the ever-growing demands for specialized models that can cater to domain-specific nuances. In our preliminary study, we find that existing models such as Alpaca (Taori et al., 2023) and Dromedary (Sun et al., 2023), although aligned, exhibit only a modest degree of improvement (compared to source models before alignment) within the specialized domains. These observations underline the need for innovative approaches that can harness the depth of domain-specific knowledge existing in the base models, to ensure the self-generated instruction-tuning data remains both contextually appropriate and accurate.

In this work, we introduce and explore the new concept of **self-specialization** (Fig. 1). Drawing inspiration from the foundational principles of self-alignment, self-specialization goes a step further by incorporating domain-specific seeds and external knowledge. Our approach integrates specialized seed instructions and is further bolstered by a retrieval component. Our goal is to guide models beyond generic alignment, directing them to generate data that are not just contextually fitting for a specialized domain but also maintain high degrees of accuracy.

Through rigorous experiments, we evaluate our self-specialized models within the biomedical domain. Surprisingly, despite the apparent simplicity of our approach, our results, present a compelling case for self-specialization. The experimental results on both biomedical and financial domains demonstrate that our self-specialized model using this approach outperforms its base model by a large margin. Notably, our self-specialized one based on MPT-30B (Team, 2023) for biomedicine even surpasses larger models (based on LLaMA-65B (Touvron et al., 2023a)), including the ones improved through self-alignment by leading methods (Taori et al., 2023; Sun et al., 2023). Moreover, we show that effective self-alignment is possible with parameter-efficient finetuning techniques (PEFT (Mangrulkar et al., 2022), QLoRA (Dettmers et al., 2023) in our case). This opens an exciting opportunity for memory-efficient multi-specialized models, where we envision that the base LLM is loaded once and is "surrounded" by a set of low-memory-footprint LoRA modules, delivering all the specialization of interest (and potentially their mix (Huang et al., 2023)) at once and in-memory without re-loading (Fig. 1).

Consequently, the contributions of our work encompass:

- We conduct comprehensive benchmarking of general-purpose aligned models within a specialized domain, underscoring the intrinsic challenge of encoding vast general knowledge into a finite set of parameters, motivating the need for specialization.

- This work explores a lightweight solution, self-specialization that enables us to uncover latent expertise within LLMs with minimal supervision.

- Our experiments in a biomedical domain demonstrate the remarkable potential of self-specialization, showcasing its efficiency and practicality. The promising results, achieved with this simple scheme, open new avenues for future work in this realm.

## 2 PRELIMINARIES: BENCHMARKING EXISTING ALIGNED MODELS

To motivate our exploration of self-specialization, we first begin by addressing a fundamental question: How well do generally aligned models perform on specialized domains? While existing popular ones such as Alpaca and Dromedary have demonstrated the generalizability of following instruc-

tions in a general scenario, it remains unclear whether general alignment can also elicit expertise for a certain domain.

Investigating this, we assess the capabilities of Alpaca (Taori et al., 2023) and Dromedary (Sun et al., 2023) against a base model, LLaMA (Touvron et al., 2023a), on a collection of benchmarks within the biomedical domain. Ensuring an unbiased comparison, all models are equally fixed with 65B parameters and share the same architecture (LLaMA). We select 10 different biomedical NLP datasets, covering a diverse set of tasks to ensure a comprehensive mix of content and also to look at the cross-task generalization, the core of instruction tuning. Few-shot (k=5) settings are examined where demonstrations are tailored to each task. Note that in fact, Alpaca is not "self"-aligned in that it uses datasets generated by GPT-3.5 (Ouyang et al., 2022) following the self-instruct process (Wang et al., 2022a), unlike Dromedary which uses the same base model. Nonetheless, we also benchmark it to serve a sort of upper bound. The results are shown in Figure 2. Details of the setups are described in Section 4.1.

When benchmarked on the expert (biomedical) domain, we find that both Alpaca and Dromedary have only a slight (1.1 - 2.5%) advantage over LLaMA. While they are aligned to handle a broad set of instructions, they do not seem to effectively improve their specialized domain expertise; intuitively trading their expertise for generality given finite parameters. In light of these findings, it becomes evident that for cases where we are only interested in target domains for all our downstream tasks, there exists a substantial potential for enhancement within domain-specific tasks. This underscores the need for a model or approach, like self-specialization, that could potentially uncover specialization while maintaining cross-task generalizability with minimal supervision. Moreover, if the approach is parameter efficient, it constitutes a considerable advantage, as many specialized models can be efficiently served in memory, sharing their deployment on the same machines / GPUs.

## 3    SELF-SPECIALIZATION

In this section, we describe our method called self-specialization illustrated in Figure 3. Starting with a select set of human-crafted, domain-specific seed instructions, the base model evolves to generate synthetic instructions and corresponding input contexts intrinsic to the domain. As we progress to the response generation phase, we enhance responses with domain-centric knowledge accessed via a retrieval mechanism, retrieving instruction-related knowledge from a domain-specific, and yet unlabeled, source. Upon generation, the specialization triggering stage follows where the base model is aligned, calibrating its expertise to resonate with the target domain. Optionally, this process can be reiterated with the aligned model as a better generator.

### 3.1    SEED INSTRUCTIONS

At the outset, we harness a curated set of seed instructions $S$, consisting of a triplet $(i, c, y)$, comprised of instruction $i$, a context $c$ (e.g., passage), and a response $y$, respectively. Considering the real-world scenarios where domain-specific data are relatively harder to acquire (Bai et al., 2021), we aim to have a very minimal number of seed instructions. For example, we use only 80 seeds for the biomedical domain and 90 seeds for the financial domain. While manual annotation of seed data is an assumed prerequisite for this initial step in self-alignment, we consider those numbers to be reasonable to annotate and leverage established datasets such as Box (Parmar et al., 2022) for seed construction to fairly ensure their quality (detailed in Section 4.1). These seeds encapsulate the "spirit" of the fundamental concepts and intricacies of the targeted domain and yet are clearly insufficient to encompass the entirety of domain knowledge. We conjecture (and surprisingly demonstrate), that the domain knowledge is already residing inside the sufficiently large models generally pre-trained, yet is in a state of superposition, which does not however prevent this knowledge from being uncovered, e.g. by the means of the proposed approach. The instruction seeds are pivotal, acting as the launching pad for the model's journey into specialization.

### 3.2    DOMAIN-SPECIFIC INSTRUCTION GENERATION

With the seed instructions in place, we move to generating domain-specific instructions. While these new instructions are grounded in the initial seeds, they grow to cover a comprehensive scope of the domain. Specifically, a base model $M_{base}$, such as MPT-30B (Team, 2023) which is "large enough", is prompted to produce new combinations of $(i, c)$ given a handful of seed demonstrations which are randomly sampled from the initial seeds pool. The newly formed instructions $i$, coupled with their corresponding input contexts $c$, shape a blueprint that the model utilizes in the following stages.

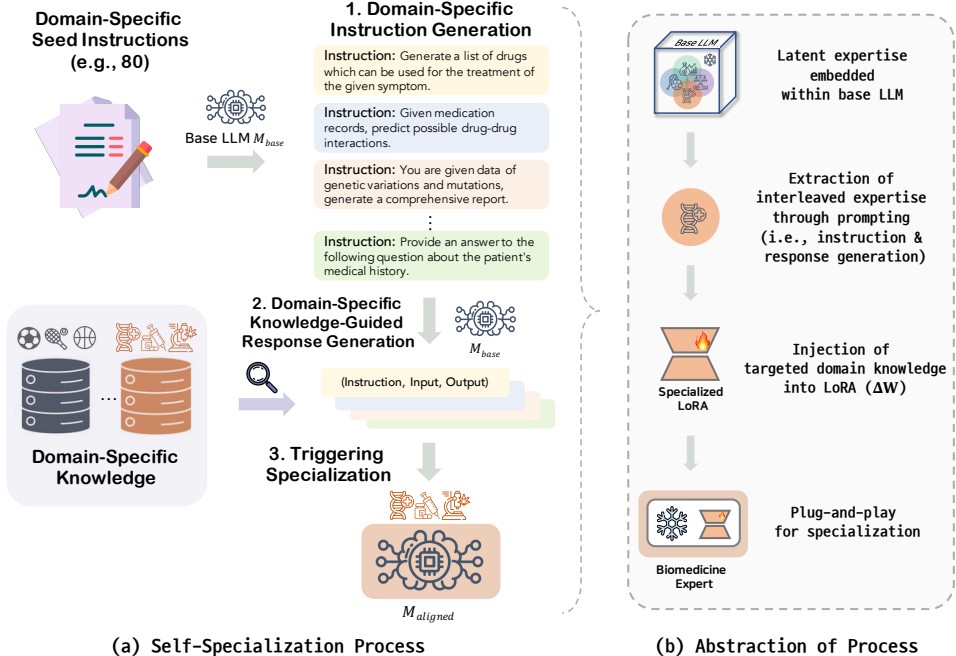

**(a) Self-Specialization Process**  **(b) Abstraction of Process**

Figure 3: An overview of **Self-Specialization**. (a) We start with a small set of human-authored domain-specific seed instructions. The base model is harnessed to craft synthetic instructions and corresponding input contexts tailored to that particular domain. Subsequently, during the response generation phase, responses are curated given the generated instruction and input pairs, enhanced by infusing domain-relevant knowledge obtained via a retrieval component. The culmination of the process is the specialization phase, where the base model undergoes specialization through tuning (w/ QLoRA) to enhance its expertise in the target domain. (b) Conceptually speaking, this process can be described as uncovering latent expertise within LLMs.

### 3.3 DOMAIN-SPECIFIC RESPONSE GENERATION

Once the synthetic domain-specific instructions $\{i\}$ and their corresponding contexts $\{c\}$ are in hand, our approach navigates to the response generation phase. It is certainly imperative for the response not only to be correct but also to be well-aligned with the target domain. We posit that incorporating external domain-relevant knowledge would be beneficial for this case, inspired by Frisoni et al. (2022). Therefore, we let the model $M_{base}$ also leverage a retrieval component, to infuse its responses with external, domain-relevant knowledge retrieved from an unlabeled domain-specific collection of documents. Specifically, forming the query $x$ as a concatenation of $i$ and $c$, a retriever $M_{ret}$ fetches top-$k$ relevant documents $d_{1:k}$.

$$d_{1:k} = M_{ret}(x = i \oplus c) \tag{1}$$

Then, each document $d_j$ is independently paired with the query $x$ to form a prompt to $M_{base}$, and the final domain-specific responses $y$ are produced from the final distribution computed by marginalizing over the probabilities of each of these $k$-combinations at each generation step. The objective of this generation can be thus formally represented as:

$$p(y|x) = \prod_i^t \sum_j^k p_{ret}(d_j|x; M_{ret}) \, p_{lm}(y_i|x, d_j, y_{1:i-1}; M_{base}) \tag{2}$$

where $p_{ret}$ is a relevance score (similarity) from a retriever module and $p_{lm}$ represents the language model distribution. By integrating such external information, while domain-specific knowledge is already deemed latent within LLMs, this step further encourages the generated target responses to be more nuanced and domain-specific, potentially leading to additional improvements (Section 4.3).

### 3.4 TRIGGERING SPECIALIZATION

Upon establishing a robust set of domain-specific responses, the model enters the specialization phase. Here, it undergoes tuning using the synthetic instructional data generated by itself, adjusting its internal parameters (i.e., QLoRA (Dettmers et al., 2023)) to cater specifically to the domain's

nuances. This step is crucial, marking the model's transformation from being generally competent to being domain-specialized while preserving cross-task generalizability, thus resulting in the final self-aligned domain-specialized model: $M_{aligned}$.

### 3.5 ITERATIVE SELF-SPECIALIZATION

In the spirit of continuous improvement, our approach optionally undergoes iterative self-specialization. It revisits the generation process of instructions and responses with the better-aligned model $M_{aligned}$. Here, to maximize the effectiveness of this process, we re-visit the idea of a contrastive decoding scheme (Li et al., 2023). The original idea of Li et al. (2023) is that a larger model can be contrasted with a smaller model in output distributions to improve the generation quality of the larger model. In our case, $M_{aligned}$ is adopted for the stronger model, whereas $M_{base}$ is considered as the weaker model, which can be represented as:

$$p(y|x) = \prod_i^t p_{lm}(y_i|x, y_{1:i-1}; M_{aligned}) - p_{lm}(y_i|x, y_{1:i-1}; M_{base}) \tag{3}$$

This process has the potential of refining the model's domain expertise with each iteration (of considering the previous iteration $M_{aligned}$ as base each time), iteratively improving its responses.

## 4 EXPERIMENTS

### 4.1 EXPERIMENTAL SETUPS

**Datasets.** For our primary evaluation, we employ various biomedical NLP datasets, most of which are curated in BIGBIO (Fries et al., 2022). A total of 10 different datasets are adopted to encompass a wide range of NLP tasks: Question Answering (QA), Named Entity Recognition (NER), Relation Extraction (RE), Sentiment Analysis (SA), and Document Classification (DC). Following a prior work (Parmar et al., 2022), all datasets are transformed into instructional data. Additionally, for the main result, we validate our method in the financial domain to showcase its generalizability. We adopt a total of 10 diverse datasets, covering numerous NLP tasks: Summarization (SUM), Question Answering (QA), Named Entity Recognition (NER), Relation Extraction (RE), Sentiment Analysis (SA), and Classification (CLS). Details on each of these datasets are elaborated in Appendix A.

**Models.** We use base MPT (Team, 2023), one of the powerful open-source foundation models, especially due to its feature of 8k context length. Inspired by the success of a previous work (Sun et al., 2023) that showed that large model size has a significant effect, we specifically adopt the 30B variant for our main experiments. For the retriever, we use simple yet effective BM25 (Robertson et al., 1994), in order to support a practical scenario where sufficient human-labeled data for training a more sophisticated retriever is not available. In addition to MPT-30B, we adopt LLaMA-2 7B (Touvron et al., 2023b) and Falcon-40B (Almazrouei et al., 2023), other strong open-source models, to further validate the general applicability of self-specialization with different scales and base models. For benchmarking of general-purpose aligned models, we evaluate Alpaca-65B (Taori et al., 2023) and Dromedary-65B (Sun et al., 2023) that are both based on LLaMA (Touvron et al., 2023a).

**Metrics.** In our study, all tasks are approached as a unified text generation problem, aiming to assess the capabilities of generative models. In alignment with an established convention (Parmar et al., 2022), we adopt $F_1$-SCORE as our main evaluation metric, given an early observation that ROUGE-L (Lin, 2004), as shown in Table 3 in Appendix, exhibits strong correlation with $F_1$-SCORE.

**Implementation Details.** For biomedical domain-specific seed data, we use data sampled from BoX (Parmar et al., 2022), which encompasses 32 tasks, up to 5 instances for each dataset, resulting in a compact yet representative seed data of 80 samples in total. These seeds are also used as demonstrations in a prompt for inference. For external corpus, we leverage PubMed[1] preprocessed in (Phan et al., 2021), which contains ≈30M abstracts. For the financial domain, we use a total of 90 seeds sampled from the 10 train sets in our corresponding benchmark datasets, and no external knowledge for simplicity. We generate 5K synthetic instructional data through the self-specialization process. Being equipped with QLoRA (Dettmers et al., 2023) and 4-bit quantization, the model is trained using a simple Alpaca-style template (Taori et al., 2023) on a single A100, taking only a few hours for 3 epochs, resulting in a light-weight (parameter efficient) specialization module that can be attached to the base model inducing its specialization upon request.

---

[1] gs://scifive/pretrain/pubmed_cleaned

Table 1: Comparative results of the base LM and self-specialized one on a biomedical domain (top) and on a financial domain (bottom). The base model is MPT-30B for biomedicine and LLaMA-2 7B for finance. Self-specialized ones have the same parameters as the counterpart base model. Performances are reported using $F_1$-SCORE. $k$ indicates the number of demonstrations in a prompt.

| BIOMEDICINE | | $k=0$ | | $k=1$ | | $k=5$ | |
|---|---|---|---|---|---|---|---|
| **Task** | **Dataset** | **Base** | **Self-Specialized** | **Base** | **Self-Specialized** | **Base** | **Self-Specialized** |
| QA | BioASQ-Factoid | 30.90 | **37.35** | 47.56 | **55.04** | 51.96 | **57.61** |
| | BioASQ-List | 46.06 | **46.99** | **47.57** | 44.55 | 35.09 | **42.17** |
| | BioASQ-Yesno | 21.20 | **85.27** | 10.80 | **94.00** | 8.80 | **95.20** |
| | PubMedQA | 11.98 | **24.16** | **28.89** | 24.87 | **31.69** | 31.31 |
| NER | AnatEM | 9.63 | **11.99** | 7.57 | **15.76** | 6.59 | **21.25** |
| | BioNLP13CG | 24.79 | **24.93** | 21.76 | **31.80** | 26.03 | **41.16** |
| | NCBI | **18.46** | 14.35 | 27.88 | **43.11** | 17.99 | **46.54** |
| RE | DDI | **51.00** | 49.40 | 49.20 | **51.60** | 49.38 | **53.40** |
| SA | Medical Drugs | 35.00 | **65.80** | 11.40 | **54.60** | 11.40 | **32.80** |
| DC | HoC | 2.44 | **6.01** | **13.91** | 7.61 | **62.84** | 62.65 |
| | Average | 25.15 | **36.63** | 26.65 | **42.29** | 30.18 | **48.41** |

| FINANCE | | $k=0$ | | $k=1$ | | $k=5$ | |
|---|---|---|---|---|---|---|---|
| **Task** | **Dataset** | **Base** | **Self-Specialized** | **Base** | **Self-Specialized** | **Base** | **Self-Specialized** |
| SUM | EDT-Summarization | 6.40 | **21.90** | 13.97 | **24.00** | 13.87 | **23.56** |
| QA | InsuranceQA | 3.03 | **19.87** | 6.55 | **23.79** | 9.96 | **24.36** |
| | ConvFinQA | **15.74** | 5.25 | **21.69** | 11.84 | **28.77** | 20.88 |
| NER | Fin3 | 9.94 | **23.93** | 7.53 | **26.95** | 6.80 | **43.87** |
| | FiNER_139 | 10.24 | **14.84** | **36.78** | 25.81 | **44.34** | 35.63 |
| RE | KPI-EDGER | 11.22 | **31.02** | 43.28 | **53.56** | 49.46 | **63.90** |
| SA | EarningsCall | 46.80 | **48.80** | **50.80** | 48.00 | **49.03** | 47.74 |
| | Financial_Phrasebank | 23.60 | **73.20** | 9.40 | **47.60** | 29.20 | **68.80** |
| | FIQA-SA | 44.44 | **56.84** | 58.55 | **61.54** | 61.54 | **70.09** |
| CLS | Gold Commodity News | 21.95 | **43.03** | **61.93** | 55.08 | 38.42 | **61.20** |
| | Average | 19.34 | **33.87** | 31.05 | **37.82** | 33.14 | **46.00** |

## 4.2 RESULTS

In Table 1, we present the comparative results of our self-specialized model against its base counterpart across 10 distinct biomedical and financial NLP tasks. MPT-30B and LLaMA-2 7B are used for biomedicine and finance, respectively. The evaluation is conducted using various $k$-shot prompting to analyze the impact of different numbers of in-context examples on model performance with/without specialization.

Our findings reveal that the self-specialized model exhibits remarkable progress in the majority of tasks across all configurations in both domains, yielding a surprisingly substantial (up to 18 points) improvement in average scores. Specifically, the scores ($F_1$) in biomedicine witness a rise from 25.15 to 36.63 in a zero-shot setting, from 26.65 to 42.29 in a 1-shot setting, and from 30.18 to 48.41 in a 5-shot setting, respectively. For finance, the gaps appear to be 14.53 (0-shot), 6.77 (1-shot), and 12.86 (5-shot), respectively. Importantly, the effectiveness of self-specialization becomes evident as it uncovers the latent expertise encoded within the "generalist" base model, showcasing the potential of leveraging inherent knowledge for enhanced domain-specific performance. These advancements in both domains underscore the self-specialized model's versatility and adaptability in addressing a wide array of tasks present in specialized domains.

**How does it compare against larger/generally aligned models?** In Figure 4, we compare our self-specialized MPT-30B model with 65B models, including LLaMA-65B, and its general instructions aligned variants in the biomedical domain. Interestingly, the results reveal that our model, despite its ≈2.2x smaller size, surpasses all 65B models. This not only highlights the lower expert domain performance trade-offs of the "generalist" models in terms of encoding a vast array of general knowledge into a finite set of parameters but also underscores the effectiveness of our (data and parameter efficient) approach to model specialization. Moreover, the efficiency and practicality of

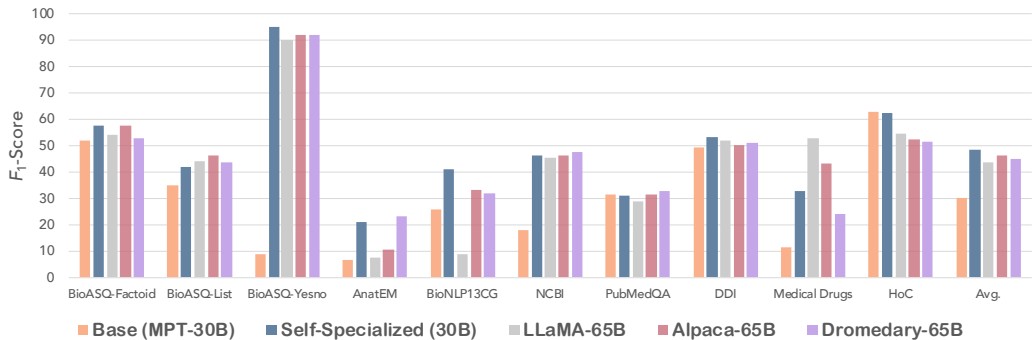

Figure 4: Results of our self-specialized model based on MPT-30B compared to 65B models in biomedicine. 5-shot results using $F_1$-SCORE are presented.

Table 2: Ablation results on iterative self-specialization and on the contribution of retrieval from unlabeled domain-specific sources during self-specialization for biomedicine. Zero-shot ($k = 0$) prompting is used for inference. Average performance over 10 tasks is reported.

|          | **Model**                      | $F_1$-SCORE | ROUGE-L |
|----------|--------------------------------|-------------|---------|
| 2nd Iter. | Self-Specialized MPT-30B      | **36.63**   | **34.79** |
|          | w/ Same Instruction Set        | 35.82       | 34.20   |
| 1st Iter. | Top-5 Docs                    | 34.57       | 32.88   |
|          | Top-1 Docs                     | 29.65       | 27.90   |
|          | No Docs                        | 33.72       | 32.14   |
|          | Base MPT-30B                   | 25.15       | 23.75   |

our simple self-specialization are further reinforced by the fact that the model is trained using only 5K[2] instruction data self-produced with minimal (only 80[3]) seeds. This training process, facilitated by the incorporation of QLoRA, which adds only 0.14% trainable parameters to an otherwise frozen model, only takes a few hours on a single (A100 80GB) GPU.

## 4.3 ABLATIONS & ANALYSES

**Effect of external knowledge.** We investigate the influence of incorporating domain-specific corpus like PubMed in the response generation phase, which enriches the model with pertinent biomedical information. As observed in the "1st Iter." section of Table 2, there is a notable variation in performance depending on the number of documents incorporated. Our findings indicate that the use of the top-5 documents yields the best results. Interestingly, incorporating only the top-1 document appears to degrade the performance, a phenomenon we conjecture is due to the noise originating from an imperfect retriever. Conversely, employing top-5 documents with probability marginalization (eq. 2) seems to mitigate this issue, enabling the model to exploit informative knowledge.

**Effect of iterative self-specialization.** In Section 3.5, we discussed the potential of employing an iterative process by leveraging the self-specialized model instead of the base model throughout the generation process. As evidenced in Table 2, initiating a "2nd Iter." of self-specialization results in further performance enhancement. Additionally, we consider two scenarios differentiated by whether the same instruction set is used to train the self-specialized model in the first and the second iterations. Our findings show that employing a distinct set of instructions for the second iteration in response generation is more effective. This could potentially be attributed to the limitation of using a confined generated instruction set, which might limit the model's generalization capabilities.

**Can self-specialization also be applied to a different model (or model size)?** To demonstrate the applicability of self-specialization other than to the MPT model, we apply it to another open-source model, Falcon-40B. Figure 5 shows the results on five different datasets. Notably, we observe a trend of improvement analogous to that of MPT-30B when self-specialization was applied to Falcon-40B, thereby substantiating that the technique is not exclusive to the MPT model. Surprisingly, despite its larger parameter size, the Self-Specialized Falcon-40B underperforms its MPT-30B counterpart, while at the same time significantly improving upon the base Falcon-40B.

---
[2]52K for Alpaca and 360K for Dromedary
[3]175 for Alpaca and 195 for Dromedary

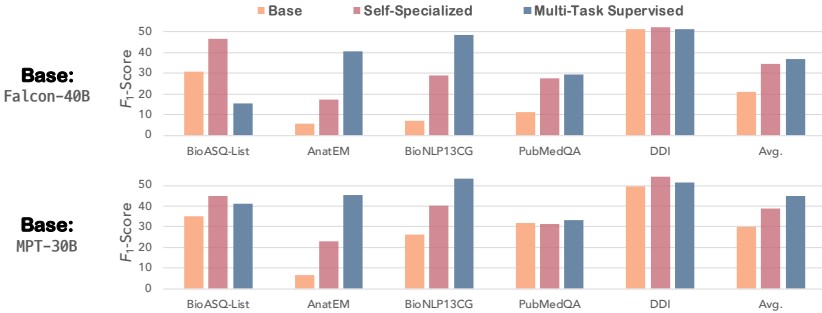

Figure 5: 5-shot results based on Falcon-40B and MPT-30B, showcasing the self-specialization gains. "Multi-Task Supervised" is a model trained on a large amount of human-labeled data in a multi-task setting and is provided *for reference* as a (non-data-efficient, expensive) *upper bound*.

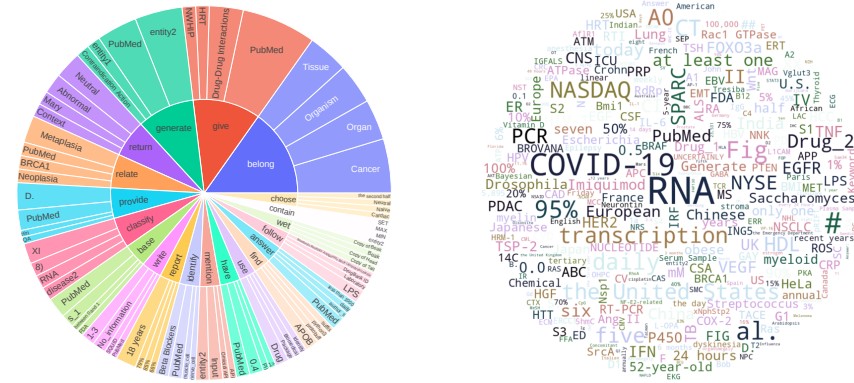

Figure 6: Statistics for instructions (left) and input context (right) generated through self-specialization. On the left, the inner circle illustrates prevalent verbs in the instructions, with the outer ring revealing associated entities. Conversely, the right side showcases the input context, highlighting the incorporation of diverse biomedical keywords. Best viewed in zoom and color.

**How is the quality of synthetic self-specialization data?** To quantitatively assess the quality of the data generated through self-specialization, we train a model using 3.7K instances of available human-labeled data in a multi-task learning setting and compare its performance to that of a model trained on 5K instances of generated synthetic self-specialization data, as depicted in Figure 5. Although the model trained on supervised data exhibits higher performance as expected, the performance gap between the two models is not large, further underscoring the effectiveness of the proposed self-specialization. In Figure 6, we showcase a qualitative visualization that analyzes the synthetic data generated through self-specialization. Additionally, some examples are provided in Table 6 & 7 in Appendix, offering insights into the quality of the self-generated specialization data.

## 5 RELATED WORK

The goal of instruction-tuning and alignment of large language models (LLMs) is to achieve cross-task generalization or to align with human preferences. This can be accomplished by either training LLMs directly with human-labeled data (Ouyang et al., 2022; Wei et al., 2022; Mishra et al., 2022; Wang et al., 2022b) or data generated by larger models (i.e., distillation) (Taori et al., 2023; Chiang et al., 2023). Recent studies have shown that LLMs are self-instructors. Wang et al. (2022a) showed that with in-context prompts, GPT-3 (Brown et al., 2020) can generate high-quality instruction-responses pairs for its own alignment. Sun et al. (2023) further suggests that using principles can minimize human supervision while covering a broad spectrum of scenarios with the open-source model, LLaMA-65B (Touvron et al., 2023a). While enhancing general alignment, according to our presented evidence, these approaches are unlikely to induce specialization in expert domains, leaving different domain expertise in "superposition" inside the model. To the best of our knowledge, we are the first to show the potential techniques for expert domain specialization through self-alignment, effectively "uncovering" a domain expert out of the model in a parameter- and data-efficient manner.

Recent studies highlight the benefits of employing instructions in different adaptation scenarios (Parmar et al., 2022). INSTRUCTOR (Su et al., 2022) illustrated the adaptability of instruction-based text embeddings to various tasks and domains, while INSTRUCTE (Bai et al., 2023) demonstrated that incorporating instructions with a schema can yield robust results for table extraction across diverse domains. However, these require the use of costly human labels or extensively tuned large models (e.g., 175B). Self-training has also been explored for different adaptation scenarios. For domain knowledge adapation, Shakeri et al. (2020) and Luo et al. (2022) proposed constructing synthetic data by generating in-domain question-answering data, but these data generators are trained with more than 80k human curated QA pairs and do not involve instructional ones that have the potential for cross-task generalization. Instruction-tuning has been shown to adapt pre-trained LLMs to different modalities, including vision (Liu et al., 2023), audio (Gong et al., 2023), and programs (Rozière et al., 2023), and enables the use of APIs (Schick et al., 2023) and search engines (Luo et al., 2023). Unlike these works, our work focuses on uncovering target domain expertise latent within LLMs while promoting cross-task generalization with minimal supervision.

## 6 DISCUSSION

While our study provides encouraging insights into the capabilities of self-specialization, this is an initial step in opening up new opportunities. We recognize that there is much to learn and explore in this exciting direction as discussed below. The promising results, achieved even with the proposed simple scheme, suggest that further refinement of this approach and exploration across diverse specialized domains could be pivotal, contributing to the ongoing efforts to uncover the embedded expertise of LLMs. In what follows, we discuss noteworthy considerations and potential directions.

**On the scale of a base model and data.** Our study primarily focuses on employing a large base model size (i.e., 30B), motivated by the success of preceding research (Wang et al., 2022a; Sun et al., 2023) that employed general self-alignment with even larger models (e.g., 65B, 175B). Nonetheless, we believe the investigation of a smaller-scale model (e.g., 7B) is a more challenging yet valuable endeavor due to its relatively limited knowledge/capability. Our exploration on such a scale in Table 1 and Figure 7 in Appendix C demonstrates the practical feasibility. With the self-specialized 30B model surpassing the performance of a 65B base model by a large margin, we are hopeful about the potential applicability of self-specialization to larger scales, though improvements could possibly be marginal by nature due to higher baseline performance. Regarding the quantity of synthetic data generated through self-specialization, we constrained it to 5K for the sake of simplicity and efficiency (Zhou et al., 2023). While this led us to highlight the data efficiency of our model, future studies could investigate the extent to which increasing the data could further enhance the expertise.

**Mixture-of-self-aligned-experts.** The potential demonstrated by self-specialization in our study paves the way for another fascinating research direction: the combination of distinct self-specialized models. This would involve integrating the expertise of various self-specialized models, aiming to create a more comprehensive and adaptable solution. Investigating the synergies and challenges in combining different specialized models (e.g., lightweight specialized LoRA layers) can lead to the development of a model with enhanced concurrent proficiency across a broader spectrum of specialized domains (at once), offering a holistic approach to maximizing the extraction and utilization of the latent expertise of LLMs.

## 7 CONCLUSION

Our exploration into self-specialization, drawing inspiration from the recent achievements in general-purpose self-alignment (Wang et al., 2022a; Sun et al., 2023), aimed to elucidate the latent expertise within large language models (LLMs) with very limited human supervision. This scheme, which incorporates a few domain-specific seeds and external knowledge into the synthetic data generation process, demonstrated promising results in a specialized domain. The self-specialized model (30B) exhibited remarkable performance, outshining its base model, MPT-30B, and even surpassing larger existing generally aligned models (65B). This illuminates the intrinsic challenges of encoding vast general knowledge into limited parameters and underscores the efficiency of self-specialization. Remarkably, the model's efficient training, marked by minimal data usage and the integration of QLoRA (Dettmers et al., 2023), adds another layer to its practicality in terms of parameter and data efficiency. These findings signify a promising advancement in the field, suggesting a pathway for leveraging inherent domain-specific expertise in LLMs and offering a large variety of exciting opportunities for future work in self-specialization.

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

## A    EXPLANATIONS OF EVALUATION DATASETS

Below are brief descriptions for each dataset in biomedical and financial domains.

### A.1    BIOMEDICINE

**BIOASQ-8B (NENTIDIS ET AL., 2020).**    This is a biomedical QA dataset that necessitates models to produce answers from given questions and corresponding contexts within the biomedical domain. There are three distinct subsets that can be divided according to question types: Factoid, List, and Yesno.

**PUBMEDQA-LONG (JIN ET AL., 2019).**    PubMedQA is another biomedical QA dataset featuring research questions along with their corresponding abstracts and answers sourced from PubMed[4]. To diversify the task types, we focus on a long-form answer (i.e., conclusion).

**ANATEM (PYYSALO & ANANIADOU, 2013).**    This is a Named Entity Recognition (NER) task for anatomical entities in biomedical texts. Models are tasked with identifying all anatomy-named entities and their corresponding types from given a small paragraph.

**BIONLP13CG (PYYSALO ET AL., 2013).**    The Cancer Genetics (CG) is an information extraction task targeting the recognition of events in text, encompassing multiple levels of biological organization, from molecular to whole organisms.

**NCBI (DOGAN ET AL., 2014).**    The NCBI disease corpus, derived from the National Center for Biotechnology Information, focuses on disease name recognition.

**DDI (HERRERO-ZAZO ET AL., 2013).**    The Drug-Drug Interaction (DDI) dataset is tailored for identifying interactions between different drugs in biomedical texts. Following Parmar et al. (2022), this work considers only binary Relation Extraction (RE), determining whether there is an effect of given two drugs.

**MEDICAL DRUGS (KHAN, 2019).**    This is a Sentiment Analysis (SA) dataset that is required to predict the sentiment of individuals towards medical drugs. Specifically, given a text and a drug, a model determines the effect of the drug as "positive", "negative", or "neutral".

**HOC (BAKER ET AL., 2015).**    The Hallmarks of Cancer (HoC) dataset is curated for classifying (zero to many) biomedical texts related to cancer into categories representing different hallmarks of cancer. In particular, these hallmarks include "sustaining proliferative signaling", "resisting cell death", "genomic instability and mutation", "activating invasion and metastasis", "tumor promoting inflammation", "evading growth suppressors", "inducing angiogenesis", "enabling replicative immortality", "avoiding immune destruction" and "cellular energetics".

### A.2    FINANCE

**EDT-SUMMARIZATION (ZHOU ET AL., 2021).**    This dataset challenges models to perform abstractive summarization on financial news articles, condensing detailed information into succinct summaries.

**INSURANCEQA (FENG ET AL., 2015).**    This is an open-book question-answering task about insurance, demanding models to extract and provide specific insurance-related information.

**CONVFINQA (CHEN ET AL., 2022).**    This is a dataset for conversational question-answering over financial report tables, testing a model's ability to reason and respond within a conversational context.

---

[4]https://www.ncbi.nlm.nih.gov/pubmed

FIN3 (SALINAS ALVARADO ET AL., 2015). This is a financial NER dataset based on financial agreements to aid credit risk assessments.

FINER_139 (LOUKAS ET AL., 2022). This NER task focuses on financial texts, where models identify and classify financial-related entities like numbers. This dataset includes a much larger label set of 139 entity types.

KPI-EDGAR (DEUSSER ET AL., 2022). Models are tasked with extracting key performance indicators (KPIs) from financial documents. Categories for KPIs include current and previous year values, annual changes, subordinate and descriptive attributes, co-references, and false-positive.

EARNINGSCALL (ROOZEN & LELLI, 2021). This is a binary sentiment analysis task where models evaluate sentiments from stock values and transcripts of earnings calls, reflecting the financial sentiments expressed.

FINANCIAL_PHRASEBANK (MALO ET AL., 2014). This dataset involves (3-way) sentiment analysis of financial news headlines, assessing the underlying sentiment conveyed by the language used.

FIQA-SA (MAIA ET AL., 2018). It consists of aspect-based sentiment analysis tasks within financial texts, requiring models to discern sentiment regarding specific aspects mentioned.

GOLD COMMODITY NEWS (SINHA & KHANDAIT, 2021). This dataset involves classifying financial news headlines about gold commodities into categories such as market movement direction or type of financial news (e.g., direction up, down, pastprice, futurenews, etc).

## B DETAILS OF EXPERIMENTS

In Table 5, we show the prompts used for our self-specialization. For instruction generation, we leverage the prompt designed in self-instruct Wang et al. (2022a) with minimal change to make it suit to specialization. In particular, we ask a model for instructions about a targeted domain, and force it to generate input together with the instruction, unlike in Wang et al. (2022a) that generates those separately. In addition, we avoid using the specific requirement in the prompt that asks to cover diverse topics, such as (quoting Wang et al. (2022a)) "daily routines, travel and tourism health and wellness, cooking and recipes, personal finance, environmental issues, history and historical events, literature and literary analysis, politics and current events, psychology and mental health, art and design, mathematics and problem-solving, physics and astronomy, biology and life sciences, chemistry and materials science, computer science and programming, engineering and technology, robotics and artificial intelligence, economics and business management, philosophy and ethics, and more". For response generation, we use a simple prompt to let a model answer with a target domain in mind. Both prompts can be further enhanced and optimized for better self-specialization performance in future work.

Regarding our evaluations, we use prompt templates that were designed and used to optimize each Alpaca (Taori et al., 2023) and Dromedary (Sun et al., 2023), but no specific template for base models, as they were not optimized for it during pre-training. Ours employs a simple Alpaca template for training and evaluation. We leverage publicly available delta weights that are supposed to be attached to LLaMA (Touvron et al., 2023a) for Dromedary, and use the ones reproduced for Alpaca in our work.

We use three seed demonstrations in-context, which are randomly sampled from our initial seeds, and sampling with top-p being 0.98 and temperature being 1.0 during instruction generation. For response generation, we use no demonstrations in-context since there is a high chance that the generated instruction task and the sampled one do not match well. We believe further exploration of this aspect would be valuable in future work. For fine-tuning, we use a batch size of 32, a learning rate of 3e-4, and epochs of 3. Low-rank adaptation (LoRA) (Hu et al., 2022; Dettmers et al., 2023) is applied to all modules and all layers with a rank of 8, and an alpha of 16.

Table 3: Comparative results (ROUGE-L) of the base LM (MPT-30B) and self-specialized one (30B) on a biomedical domain. $k$ indicates the number of demonstrations in a prompt. ROUGE-L exhibits the same trend with $F_1$-SCORE .

| BIOMEDICINE | | k=0 | | k=1 | | k=5 | |
| --- | --- | --- | --- | --- | --- | --- | --- |
| Task | Dataset | Base | Self-Specialized | Base | Self-Specialized | Base | Self-Specialized |
| QA | BioASQ-Factoid | 30.70 | **37.31** | 47.35 | **54.71** | 51.81 | **57.48** |
| | BioASQ-List | 41.07 | **40.65** | **42.38** | 38.50 | 30.40 | **36.24** |
| | BioASQ-Yesno | 21.20 | **85.27** | 10.80 | **94.00** | 8.80 | **95.20** |
| | PubMedQA | 9.15 | **18.88** | 22.78 | 18.52 | 24.56 | **24.77** |
| NER | AnatEM | 8.65 | **10.69** | 6.67 | **13.83** | 6.07 | **19.24** |
| | BioNLP13CG | **20.41** | 20.34 | 19.02 | **27.54** | 22.53 | **35.07** |
| | NCBI | **17.94** | 13.75 | 25.22 | **39.27** | 16.60 | **41.55** |
| RE | DDI | **51.00** | 49.40 | 49.20 | **51.60** | 49.38 | **53.40** |
| SA | Medical Drugs | 35.00 | **65.80** | 11.40 | **54.60** | 11.40 | **32.80** |
| DC | HoC | 2.42 | **5.83** | 13.88 | 7.61 | **62.84** | 62.61 |
| Average | | 23.75 | **34.79** | 24.87 | **40.02** | 28.44 | **45.84** |

# C  ADDITIOANL RESULTS

**Qualitative Analyses.**   While our study primarily focuses on the biomedical and finance domain, the applicability and effectiveness of self-specialization in another specialized domain whose knowledge is relatively limited, such as sports, remain an open avenue for exploration. As an initial effort, we present a case study of a self-specialized model on sports in Table 8 & 9, along with the visualization of generated data in Figure 8. We hope that this could offer insights into the versatility of self-specialization, although the model is not yet perfect, and thorough evaluations are required in future work. Different domains inherently pose unique requirements and nuances, and understanding how self-specialization adapts to these variations is a valuable direction for future work.

**On the Sensitivity of Prompting.**   In Table 1, we observe the decreased performances with increased demonstrations in certain cases such as Medical Drugs. We conjecture this can be attributed to the model's sensitivity to the provided context in the few-shot setting. With specialized instruction and more demonstrations, the model receives more information, which can sometimes lead to confusion and a decline in performance, especially if the resulting prompt is complex and lengthy (e.g., 4K). Taking the worst average, and the best across different k-shot (0, 1, 5) configurations for each dataset to address the concern of sensitivity, we still notice the significant gaps between our self-specialization and the base model, presented in Table 4.

**Self-Specialization with a Smaller Scale and Comparison with Existing Baselines.**   We investigate self-specialization with a smaller scale 7B model for the biomedical domain, which is deemed a more challenging yet insightful endeavor due to its relatively limited knowledge/capability. As shown in Figure 7, the findings validate the efficacy of self-specialization even at this scale. Furthermore, we compare our model with existing baselines (Wu et al., 2023): MedLLaMA-13B and PMC-LLaMA-7B/-13B. MedLLaMA is a LLaMA variant further pre-trained on a huge domain-specific corpus (i.e., medicine), and PMC-LLaMA is further instruction-tuned using existing annotated datasets as well as synthetic datasets, encompassing medical question-answering (QA), rationale for reasoning, and conversational dialogues. Notably, we find that our self-specialized 7B model is on par with or better than both MedLLaMA-13B and PMC-LLaMA-13B despite their larger parameters and extensive domain-specific training. This further emphasizes the effectiveness of our approach. Additionally, using our 7B-generated data to specialize MedLLaMA indicates that self-specialization can enhance domain-specific pre-training, suggesting complementarity.

Table 4: Comparative results of the base LM and self-specialized one on a biomedical domain (top) and on a financial domain (bottom). The base model is MPT-30B for biomedicine and LLaMA-2 7B for finance. Self-specialized ones have the same parameters as the counterpart base model. Performances are reported using $F_1$-SCORE. The results are presented using worst, average, and best across 0-, 1-, and 5-shot results for each dataset.

| BIOMEDICINE | | Worst | | Average | | Best | |
|---|---|---|---|---|---|---|---|
| Task | Dataset | Base | Self-Specialized | Base | Self-Specialized | Base | Self-Specialized |
| QA | BioASQ-Factoid | 30.90 | **37.35** | 43.47 | **50.00** | 51.96 | **57.61** |
| | BioASQ-List | 35.09 | **42.17** | 42.91 | **44.57** | **47.57** | 46.99 |
| | BioASQ-Yesno | 8.80 | **85.27** | 13.60 | **91.49** | 21.20 | **95.20** |
| | PubMedQA | 11.98 | **24.16** | 24.19 | **26.78** | **31.69** | 31.31 |
| NER | AnatEM | 6.59 | **11.99** | 7.93 | **16.33** | 9.63 | **21.25** |
| | BioNLP13CG | 21.76 | **24.93** | 24.19 | **32.63** | 26.03 | **41.16** |
| | NCBI | **17.99** | 14.35 | 21.44 | **34.67** | 27.88 | **46.54** |
| RE | DDI | **49.20** | 49.40 | 49.86 | **51.47** | 51.00 | **53.40** |
| SA | Medical Drugs | 11.40 | **32.80** | 19.27 | **51.07** | 35.00 | **65.80** |
| DC | HoC | 2.44 | **6.01** | 26.40 | 25.42 | 62.84 | 62.65 |
| | Average | 19.62 | **32.84** | 27.33 | **42.44** | 36.48 | **52.19** |

| FINANCE | | Worst | | Average | | Best | |
|---|---|---|---|---|---|---|---|
| Task | Dataset | Base | Self-Specialized | Base | Self-Specialized | Base | Self-Specialized |
| SUM | EDT-Summarization | 6.40 | **21.90** | 11.41 | **23.15** | 13.97 | **24.00** |
| QA | InsuranceQA | 3.03 | **19.87** | 6.51 | **22.67** | 9.96 | **24.36** |
| | ConvFinQA | **15.74** | 5.25 | **22.07** | 12.66 | **28.77** | 20.88 |
| NER | Fin3 | 6.80 | **23.93** | 8.09 | **31.58** | 9.94 | **43.87** |
| | FiNER_139 | 10.24 | **14.84** | 30.45 | 25.43 | **44.34** | 35.63 |
| RE | KPI-EDGER | 11.22 | **31.02** | 34.65 | **49.49** | 49.46 | **63.90** |
| SA | EarningsCall | 46.80 | **47.74** | **48.88** | 48.18 | **50.08** | 48.80 |
| | Financial_Phrasebank | 9.4 | **47.60** | 20.73 | **63.20** | 29.20 | **73.20** |
| | FIQA-SA | 44.44 | **56.84** | 54.84 | **62.82** | 61.54 | **70.09** |
| CLS | Gold Commodity News | 21.95 | **43.03** | 40.77 | **53.10** | 61.93 | 61.20 |
| | Average | 17.60 | **31.20** | 27.84 | **39.23** | 35.99 | **46.59** |

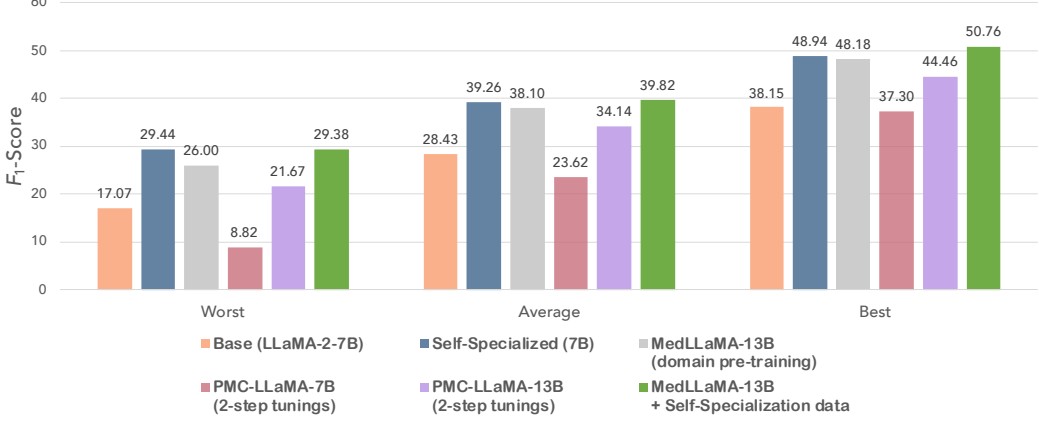

Figure 7: Results in biomedicine using LLaMA-2 7B as a base model, and comparisons with other baselines including the one pre-trained on a huge domain-specific corpus. The results are presented using worst, average, and best across 0-, 1-, and 5-shot results for each dataset.

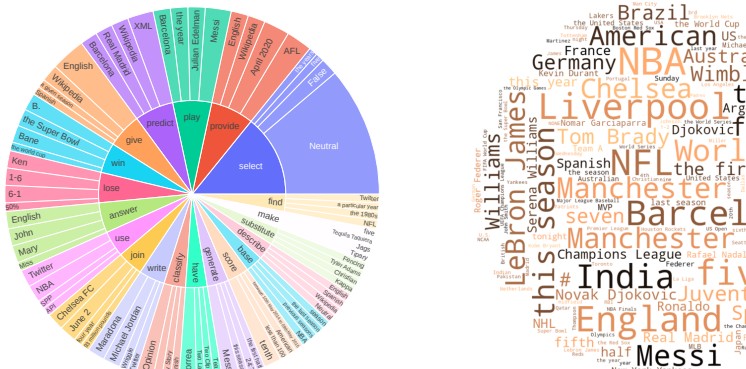

Figure 8: Statistics for instructions (left) and input context (right) generated through self-specialization toward the sports domain, with 40 seeds, 1st iteration only, and no retrieval component. On the left, the inner circle illustrates prevalent verbs in the instructions, with the outer ring revealing associated entities. Conversely, the right side showcases the input context, highlighting the diverse sports keywords generated by the model in the process of self-specialization. Best viewed in zoom and color.

Table 5: Prompts used for instruction generation and response generation.

### Instruction Generation Prompt

```
You are asked to come up with a set of 20 diverse task instructions about a biomedical domain.
These task instructions will be given to a GPT model and we will evaluate the GPT model for
completing the instructions.

Here are the requirements:
1.  Try not to repeat the verb for each instruction to maximize diversity.
2.  The language used for the instruction also should be diverse.  For example, you should
combine questions with imperative instructions.
3.  The type of instructions should be diverse.  The list should include diverse types of
tasks like open-ended generation, classification, editing, etc.
4.  A GPT language model should be able to complete the instruction.  For example, do not
ask the assistant to create any visual or audio output.  For another example, do not ask the
assistant to wake you up at 5pm or set a reminder because it cannot perform any action.
5.  The instructions should be in English.
6.  The instructions should be 1 to 2 sentences long.  Either an imperative sentence or a
question is permitted.
7.  You should generate an appropriate input to the instruction.  The input field should
contain a specific example provided for the instruction.  It should involve realistic data and
should not contain simple placeholders.  The input should provide substantial content to make
the instruction challenging.
8.  Ensure diverse tasks are covered in the instructions and inputs, while focusing on a
biomedical domain.

List of 20 tasks:
```

### Response Generation Prompt

```
You are a biomedical domain expert.  Given an instruction and an input, generate the best
response to solve the given biomedical task.
```

Table 6: A list of examples (#1) of generated data on biomedicine through self-specialization.

---

*Instruction*
The aim of this task is to identify the functional categories and chemical categories
associated with a data record. The record is given in a pre-defined form and followed by
a set of supporting sentences that might be relevant. In particular, for each data record,
a triplet of functional categories, molecular categories and chemical categories associated
with the data record will be provided. The task is to find some relations between records and
supporting evidence that can support the given relations.

*Input*
Data record: Atom, CNA, NOC, NFT
Text: Cisplatin and carboplatin, two drugs that are used to treat testicular and ovarian
cancers, are not effective in treating breast cancer

*Output*
The functional category of cisplatin is CNA. The chemical category of cisplatin is NOC. The
molecular category of cisplatin is NFT.

---

*Instruction*
In this task, you are given a short article and question. Read the short article and answer
the question.

*Input*
Short article: The extract from the opium poppy was tested on breast cancer cells and was
found to inhibit the migration and invasion of breast cancer cells.
Question: Was the extract more potent in its inhibitory effect on the migration of breast
cancer cells than its effect on the invasion of breast cancer cells?

*Output*
Yes.

---

*Instruction*
In this task, you are given a paragraph, and your task is to categorize the entities from the
given input into two types: Dyes and Protein Molecules. The defined types here is based on
the medical specialty such that dyes are typically for understanding human metabolism.

*Input*
Paragraph: Thus in an effort to improve the characterization of such compounds an alternative
glycosylation labeling strategy was implemented based on malonate attachment to glucosamine
and sond glycosyltransferase inactivation.

*Output*
Dyes: malonate, Protein Molecules: glucosamine, sond glycosyltransferase.

---

*Instruction*
In this task, you are given a small paragraph, your task is to identify the relationship
between the variables. Find the most likely to be correct relationship of a gene with its
characteristics. For example, if one entity is pathway, then you need to find the type
of relationship that specific gene is related to that pathway. Give the most appropriate
relationship type and tokens separating two variables. Given an input, determine the
relationship type.

*Input*
COPD is a major cause of morbidity and mortality throughout the world, with over three million
deaths per year Pathway: respiratory system Pathway: stress Hsa-miR-30b-5p is a regulator of
retinal ganglion cell apoptosis through regulating the PTEN/AKT signaling pathway in glaucoma.

*Output*
Hsa-miR-30b-5p is a regulator of retinal ganglion cell apoptosis through regulating the
PTEN/AKT signaling pathway in glaucoma.

---

*Instruction*
In this task, you are given a question and a sentence as context. You need to find the
answers to the question in the given context. You can either answer the question or ask a
question. Given the question, a context or a question, this task must either answer the
question or ask a question.

*Input*
Question: For what drugs can predict the effect of changing the dose
Context: Metoprolol is a commonly used beta-blocker, which competitively blocks
beta-adrenergic receptors in the myocardium. Adrenergic beta-blockers do not antagonize
the presynaptic beta-1 receptors in the conduction system. While they thus prolong
the ventricular conduction time, they do not alter conduction velocity. Metoprolol is
therapeutically active over the heart rate range of 40 to 100 bpm, and may be particularly
effective for the rate control of atrial fibrillation. The beta-blocking action of metoprolol
is accompanied by an increase in stroke volume.

*Output*
The beta-blocking action of metoprolol is accompanied by an increase in stroke volume.

---

Table 7: A list of examples (#2) of generated data on biomedicine through self-specialization.

---

*Instruction*
In this task, you are given a drug name, and the side effect caused due to this drug (e.g., nausea). Your job is to generate other side effects in which this drug often causes.

*Input*
Drug_1: Phentermine Drug_2: Nausea

*Output*
Headache, insomnia, constipation, dry mouth, diarrhea, anorexia, increased blood pressure, increased heart rate, increased appetite, dizziness, nervousness, restlessness, irritability, tremor, rash, itching, and urticaria.

---

*Instruction*
Given a premise and a hypothesis sentence, your task is to identify if the premise supports the hypothesis or the premise contradicts the hypothesis sentence. Your task is to identify if the evidence in the premise supports or contradicts the hypothesis. There is no other alternative. You can select two labels for this task: Support or Contradict.

*Input*
Premise: If too much calcium enters the sea, if causes it to become more acidic in a process called "ocean acidification". Hypothesis: The ocean will become acidic.

*Output*
The vitamin did not seem to work for malaria. It was noted that the protein has is able to contact the insulin-like growth factor - 1 receptor, so an antagonist was made using a modified version of the protein.

---

*Instruction*
In this task, given the title of scientific study, your task is to generate a paper abstract or summary based on the given information.

*Input*
Title of study "Type and impact of end-user requests for application programming interfaces in three health-related internet services"

*Output*
The study found that the majority of end-user requests for application programming interfaces (APIs) in three health-related internet services were for data access. The study also found that the most common type of request was for data access, followed by data analysis and data visualization. The study also found that the most common impact

---

*Instruction*
In this task, you are given a question as input, and your task is to give answer it by generating a list of drugs which can be used for the treatment of the symptom. If the symptom is rare, the answer list should contain its corresponding differential diagnosis. Given a question related to symptoms, generate a list of drugs and differential diagnosis from open vocabulary.

*Input*
Question: What are the signs and symptoms of Leptospirosis?

*Output*
Fever, headache, muscle aches, nausea, vomiting, abdominal pain, diarrhea, cough, and red eyes. Differential diagnosis: Leptospirosis, Lyme disease, Rocky Mountain spotted fever, and dengue fever.

---

*Instruction*
In this task, you are given a list of paragraphs. The list contains sentences related to a particular biomedical concept and sentences not related to a particular biomedical concept. Classify these sentences as true or false. Answer if the sentences in the input are true or false. True indicates that all the sentences of the list are true for the given biomedical concept. False indicates that only one of the given sentences is false, while the rest of them are true.

*Input*
Some of the complications associated with developing small molecules for FGFR1 and KRAS targetable therapeutic compounds are revealed as the targets have low single amino acid sequence identity between species and both are, at least partially, regulated by post-translational modifications. S. Kisor, S. Khankorn, "Drug discovery from big data: past, present, and future.", Outline, January 2013.

*Output*
False

---

Table 8: Case study on sports domain. Self-Specialized model produces better responses given the same instruction about sports.

| Model | Base MPT-30B | Self-Specialized MPT-30B |
|---|---|---|
| **Instruction** | Given a live commentary for a sports game, generate plausible sports news highlights. | |
| **Input Context** | Commentary
1. Full-time at the Etihad. Manchester City JUST manage to claim all three points in this Premier League clash as they overcome Swansea with a narrow 2-1 scoreline. The man of the match was Gabriel Jesus, who netted two goals, including the dramatic winner in the 92nd minute, which will prove to be a massive blow for the Swans.
2. GOOOOAAAALLLL! GOOOAAALLL FOR MANCHESTER CITY! ...
... | |
| **Response** | A: I think you should try to use a pre-trained model for the encoder and decoder.
... | News:
- Manchester City have come from behind to beat Swansea 2-1 in a thrilling Premier League clash at the Etihad Stadium.
... |
| **Instruction** | This task aims to predict the sentiment of a tweet about sports. Given a tweet about football written by a real user, select an answer (polarity) among "Positive", "Negative" and "Neutral". | |
| **Input Context** | @user1 @user2 @user3 @user4 @user5 Now, PSG is considered a weak team. Poor ignorant. When Barça lost 3-0, it was because we faced PSG without Neymar. But now that Real Madrid eliminated them, suddenly PSG is a weak team. ... ... | |
| **Response** | Positive | Negative |
| **Instruction** | In this task, you are given a sentence related to sports. Your job is to determine whether the provided sentence makes sense. You need to produce an answer as "plausible" or "implausible". | |
| **Input Context** | Lionel Messi hit a three-run homer. | |
| **Response** | Plausible | Implausible |
| **Instruction** | In this task, you are given a question and a context about sports, and you need to answer the question based on the given context. | |
| **Input Context** | Passage (Live Commentary):
As the match reaches a decisive point, Williams, the tennis star, finds herself engaged in a fierce rally. Her opponent delivers a powerful cross-court forehand that seems destined for a winner. However, Williams displays incredible reflexes and anticipation, executing a lightning-fast backhand passing shot down the line that catches her opponent off guard and secures the point.
Question: What remarkable shot did the player execute to win the crucial point?
Answer Choices:
a) Williams delivered a drop shot to surprise her opponent.
b) Williams hit a powerful forehand winner cross-court.
c) Williams executed a precise backhand passing shot.
d) Williams served an ace to win the point. | |
| **Response** | a) Williams delivered a drop shot to surprise her opponent. | c) Williams executed a precise backhand passing shot. |
| **Instruction** | In this task, you are given a question about sports. Answer the question based on your implicit knowledge. | |
| **Input Context** | What does the term "love" mean in tennis scoring? | |
| **Response** | It means that the player has won the game. | Love is a score of zero in tennis. |

Table 9: Case study on sports domain. Negative cases where both models produce wrong responses are presented.

| Model | Base MPT-30B | Self-Specialized MPT-30B |
|---|---|---|
| **Instruction** | In this task, you are given a set of attribute-value pairs about sports in the raw XML format. Generate a natural language text describing the given data. | |
| **Input Context** | `<originaltripleset>`
`<otriple>batterName \| Pujols</otriple>`
`<otriple>finalScore \| 16-7</otriple>`
`<otriple>gameTally \| 2-1</otriple>`
`<otriple>hasLostTeam \| Texas_Rangers</otriple>`
`<otriple>hasWonTeam \| St._Louis_Cardinals</otriple>`
`<otriple>hitNumber \| five</otriple>`
`<otriple>homeRunNumber \| three</otriple>`
`<otriple>matchDate \| Saturday_night</otriple>`
`<otriple>pitchResult \| homers</otriple>`
`<otriple>runNumber \| six</otriple>`
`</originaltripleset>` | |
| **Response** | Albert Pujols hit three home runs and drove in five runs as the St. Louis Cardinals beat the Texas Rangers 16-7 on Saturday night. | Pujols hit five home runs and drove in six runs as the St. Louis Cardinals beat the Texas Rangers 16-7 on Saturday night. |
| **Instruction** | In this task, you are given a sentence related to sports. Your job is to determine whether the provided sentence makes sense. You need to produce an answer as "plausible" or "implausible". | |
| **Input Context** | Mike Trout took ball four in the World Series. | |
| **Response** | Implausible | Implausible |
| **Instruction** | In this task, you are given a question and a context about sports, and you need to answer the question based on the given context. | |
| **Input Context** | Passage (Live Commentary):
With two runners on base and a full count, Johnson, the opposing team's batter, faced an intense battle against the pitcher. The tension reached its peak as the pitcher delivered a devastating curveball, catching Johnson off guard. He swung and missed, resulting in a resounding strikeout that ended the inning and stranded the runners.
Question: When did the pitcher deliver a crucial strikeout to end the inning?
Answer Choices:
a) At the start of the inning, Johnson struck out.
b) After a series of foul balls, Johnson hit a double.
c) At the end of the inning, Johnson grounded out.
d) With a full count, Johnson struck out to end the inning. | |
| **Response** | a) At the start of the inning, Johnson struck out. | c) At the end of the inning, Johnson grounded out. |

