# OpenReview forum: "Self-Specialization: Uncovering Latent Expertise within Large Language Models"
_ICLR.cc/2024/Conference — ICLR 2024 Conference Withdrawn Submission_

### Official Review · Reviewer_PQqS · 2023-10-22

**Soundness:** 3 good
**Presentation:** 4 excellent
**Contribution:** 2 fair
**Rating:** 5
**Confidence:** 5

**Summary:**

This study is driven by the observed limitations of pre-trained large language models (LLMs) when handling specialized tasks, particularly in the biomedical question and answering domain.

The underlying issues stem from the practice of fine-tuning these LLMs on broad instruction datasets, which often results in only marginal enhancements for niche domains.

To address this, the authors introduce a novel technique termed "self-specialization." Unlike the conventional self-alignment method, self-specialization incorporates domain-specific seeds and an external knowledge retriever. This strategy cultivates an instruction dataset tailored to the specific needs of the target domain.

Empirical results highlight the efficacy of this approach: LLMs that underwent self-specialization significantly outperformed their non-specialized counterparts.

**Strengths:**

- Methodology: The self-specialization technique introduced is both straightforward and powerful. The authors effectively demonstrate that, by refining the instruction dataset for a target domain, one can substantially elevate the performance of an LLM.

- Results: The outcomes are compelling. The self-specialized technique not only outpaces the baseline LLM with general instruction tuning across various NLP tasks, but even a 30B self-specialized model can occasionally surpass larger models of 65B capacity.

- Structured Presentation & Clear Communication: This research is characterized by its lucid motivation and coherent narrative, seamlessly bridging the gap between identified issues and the proposed solutions. The results are presented with clarity, reinforcing the dominance of the introduced methods.

**Weaknesses:**

At its core, the proposed method is somewhat an amalgamation of existing concepts. While termed "self-specialization", it essentially contrasts with "self-alignment" (for instance, as seen in Alpaca) and incorporates "domain-specific knowledge-guided generation". The enhancements observed are not surprising since the retrieval-augmented generation, grounded on specific knowledge retrieval, naturally offers a more thorough and targeted knowledge base for instruction tuning. Thus, one could argue that self-specialization is essentially self-alignment augmented with RAG, somewhat constraining its novelty.

Furthermore, rather than relying on RAG to generate the instruction dataset with the foundational model, the authors might have delved deeper into refining the dataset construction process. Several uncharted avenues remain: (1) devising superior methods for seed generation, considering aspects like topic ratios or instruction diversity; (2) optimizing answer generation for the instructions, with focus on enhancing and validating answer quality. Regrettably, this research offers only a cursory exploration in these dimensions.

**Questions:**

- In 4.1, the authors mentioned 5K synthetic instruction dataset is generated. Was there any reason for this number? If we increase/decrease the number, how would the yielded instruction-tuned model perform?

---

> ### Author Response · Authors · 2023-11-13
>
> Thank you very much for your thoughtful and constructive review of our work. We appreciate your positive comments on our method, results, and presentations, as well as your concerns on our contributions. Down below, we address each of your comments and questions in detail.
>
> >***At its core, the proposed method is somewhat an amalgamation of existing concepts. ...***
>
> We appreciate your valuable comments. While our method builds upon existing concepts of self-alignment and retrieval-augmentation, its unique angle lies in its efficient and practical use of the latent knowledge within LLMs to specialize them. This is a significant departure from standard general self-alignment and demonstrates a novel application in the field of domain specialization. While retrieval-augmentation is generally recognized for enhancing model capabilities at inference, our work explores its impact within a self-alignment (i.e., data generation) framework, which was unknown. In fact, one of the interesting and surprising findings we demonstrated was the effectiveness of self-specialization even without relying on the retrieval component while it does help to some extent, highlighting its simplicity and practicality.
>
> Finally, we would also like to note that our observation that language model specialization can be achieved in a parameter-efficient (QLoRA) way and by applying (our updated) self-alignment technique is new and valuable. This opens an exciting opportunity for efficient multi-specialized models, where a single base LLM is loaded once and is surrounded by a set of low-memory-footprint specialized modules in-memory, delivering all the specialization of interest without re-loading, a highly practical scenario enabled by our work. As an example, we successfully self-specialized the same LLaMA-2-7B base model to both biomed and finance domains, enabling the aforementioned efficient serving of specialization scenarios.
>
> >***Furthermore, rather than relying on RAG to generate the instruction dataset with the foundational model, the authors might have delved deeper into refining the dataset construction process. …***
>
> We thank you for your insightful suggestions regarding the refinement of the dataset construction process. Your points about exploring advanced methods for seed generation and optimizing answer generation are indeed valuable. In our current research, while we have initiated steps towards these directions, the primary focus has been on demonstrating the efficacy of the self-specialization approach itself. We believe our initial effort in this direction can serve as a foundational framework, unveiling for the first time (to the best of our knowledge) the possibility of effectively specializing models in a parameter-efficient way and without any expensive specialization domain data collection, by extracting the specialization data from the model itself. We believe our work opens interesting opportunities for further exploration and advanced refinement upon our work in future work.
>
> >***In 4.1, the authors mentioned 5K synthetic instruction dataset is generated. Was there any reason for this number? …***
>
> The selection of this number was simply a pragmatic decision, balancing computational feasibility with the need for a sufficiently diverse set of instructions for effective model specialization. We recognize that variations in dataset size could impact performance, typically with smaller datasets potentially leading to poorer results and larger ones to better results to some extent. We do think how to select the optimal number of synthetically generated data depending on target domains/tasks could be another interesting dimension to investigate, since this is challenging in low-data scenarios where automatic hyperparameter tuning might be infeasible, as also seen in previous works on self-alignment [1, 2], which relies on arbitrary numbers.
>
> [1] Wang, Yizhong et al. “Self-Instruct: Aligning Language Models with Self-Generated Instructions.” ACL 2023.
>
> [2] Sun, Zhiqing et al. “Principle-Driven Self-Alignment of Language Models from Scratch with Minimal Human Supervision.” Neurips 2023.

---

> ### Author Response · Authors · 2023-11-21
> **Looking Forward to Your Reply**
>
> Dear Reviewer PQqS,
>
> Since the discussion period is ending tomorrow, we would greatly appreciate it if you could take a look at our response to your review and let us know if you have any remaining questions. Otherwise, we would really appreciate it if you could support the paper by increasing the score. We look forward to hearing from you and addressing any remaining concerns before the end of the discussion period.
>
> Best regards,
>
> Authors

---

> > ### Comment · Reviewer_PQqS · 2023-11-22
> > **Thanksf for youre response**
> >
> > Thanks for the response from the authors. I generally agree with the empirical gain and practical value of the work. The proposed "self-alignment in domain specialization" is interesting and is important in practice. However, I would like to keep the score since there still lies the concern about the novelty and technical depth of the proposed framework. As mentioned before, the way this framework achieves "domain specialization" lies in retrieving domain knowledge and augmenting the teacher large language model, which is worth a deeper exploration. The proposed "qLoRA" scenario is interesting but is not considered as the contribution of this paper.

---

> ### Author Response · Authors · 2023-11-22
> **Thank You for Sharing Your Perspective!**
>
> Thank you for sharing your perspective! It appears there might be a misunderstanding about the role of retrieval augmentation, which we would like to clarify. In our work, the primary mechanism for achieving domain specialization in large language models (LLMs) is not predominantly reliant on retrieval augmentation. Instead, our approach focuses on uncovering and leveraging the latent expertise within LLMs themselves. The novelty of our method lies in its ability to tap into this inherent capacity of LLMs to specialize in specific domains, rather than merely the use of retrieval augmentation.
>
> In support of this, we refer you to the results presented in Table 2 of our paper. Here, we demonstrated the significant effectiveness of our self-specialization, independent of the retrieval component, where we observed an 8.5 F1 improvement compared to the base model, which is surprising. You mentioned that the enhancement through retrieval augmentation is not surprising - the reality is that the contribution of retrieval in self-specialization is relatively marginal, less than 1 point.
>
> We would like to re-emphasize that while retrieval augmentation does play a role in our framework, it serves as an additional enhancement rather than a fundamental necessity. We hope this clarification sheds light on the fundamental aspects of our approach and reassures the intended focus of our research.

---

### Official Review · Reviewer_CV1V · 2023-10-30

**Soundness:** 2 fair
**Presentation:** 3 good
**Contribution:** 2 fair
**Rating:** 3
**Confidence:** 3

**Summary:**

The paper proposes a self-specialization mechanism to automatically generate domain-specific instructional data and improve language model performance within specialized domains. The proposed approach is straightforward: 1) sample seed instructions from established NLP benchmarks within the target domain; 2) utilize the language model to generate new domain-specific instructions based on the initially sampled seeds; 3) employ domain-relevant unlabeled documents to generate responses corresponding to the generated instructions; 4) fine-tune the language model using the synthesized instructions, resulting in the development of a final domain-specialized model.

**Strengths:**

The paper is well-motivated, as specializing the language model in a specific domain is an area of interest. Additionally, the paper is well-structured and easy to follow.

**Weaknesses:**

Despite the promising results reported in this paper within the biomedical domain, I still feel uncertain about its contribution for the following reasons:
1) The experimental results are not convincing. Table 1 reports the comparative results of the base model and the self-specialized model, with the authors stating "the scores (F1) witness a rise from 25.15 to 36.63 in a zero-shot setting." However, this is the average result, and the datasets "BioASQ-Yesno" and "Medical Drugs" contribute the most differences, with a difference of around 60-70 in "BioASQ-Yesno. I highly doubt the correctness of the results for this dataset. This is a binary classification dataset, and the base model performs much worse than random guessing (50.0 if labels are balanced), which is the first odd point. Secondly, as the number of demonstrations increases, the F1 score of the base model surprisingly decreases. This implies that the base model must learn information from the prompted demonstrations. If the model is not performing well on a dataset, the F1 score should remain almost the same or should not exhibit such consistent declines. I am considering if the authors mistakenly reversed the labels of "yes" and "no," leading to these counterintuitive results.
If we exclude the "BioASQ-Yesno" and "Medical Drugs" datasets, the average F1 score improvement is down to around 2.5, far less than the reported 11.48 (36.63 - 25.15). This makes doubt not only on the reported results in Table 1 but also on Fig. 4 and its claims of the model's superiority over all 65B models despite its ≈2.2x smaller size.
2) The Gouge-L scores for the "DDI" and "Medical" datasets are identical to their respective F1 scores. I haven't investigated if this is possible, but please check the reported results carefully.
3) During the seed generation phase, the proposed method requires the NLP benchmarks in the target domain. This limits the proposed approach to extend to other domains, as not all domains have this NLP benchmark accessible. With the paper using 80 seeds, one possible solution could be manually generating them; however, the paper didn't mention this point.
4) How about the results if applying the proposed scheme to a larger model, such as LLaMA-65B? The base model used in the paper initially shows not good results (20-30 F1 scores), which are relatively easy to improve. If a larger model with better initial results is used, is it still feasible to have improvement through the proposed scheme?
5) Some places lack professional writing. For example, in Eq. (3), $p_{lm}$ is introduced without prior definition. While the intended meaning may be inferred, a scientific paper should maintain consistency and rigor in its notation and explanations.

**Questions:**

Please consider the questions I listed in the Weaknesses.

---

> ### Author Response · Authors · 2023-11-13
>
> Thank you very much for your thoughtful and constructive review of our work. We appreciate your positive comments recognizing the motivation of our work, and the quality of our paper, as well as concerns which we understand. Down below, we address each of your comments in detail.
>
> >***The experimental results are not convincing. …***
>
> We truly appreciate your critical scrutiny of our results, particularly the below-random performance observed on the “BioASQ-Yesno”, which is one out of 10 different datasets we evaluated. The nature of our evaluation is indeed the key to understanding these results. In our study, as described in Section 4.1, we treat all tasks as a unified text generation problem, aiming to assess the realistic capabilities of following instructions, consistent with established practices in biomedical instruction tuning literature [1]. This means that the model could generate responses that are not confined to “yes” or “no” labels, but rather any possible sequence of text, including those that are entirely out of the space of expected answers, not following and understanding the specialized instructions. Even if converting the minimum F1 score as 50 (random guessing), the average difference between the self-specialized and base model will be 8.6 (36.63 - 28.03), which is still significant. However, we believe that our current evaluation is fairer and preferable, because in a realistic scenario where a user prompts a model to solve a certain task (e.g., classification) without the assumption about a task type, and gets a totally wrong response out of the label space, evaluating such a response as correct would not make sense.
>
> In response to the second point about the decreased performances with increased demonstrations in 2 out of 10 datasets, after careful examination, we can assure that the labels (e.g., “yes” and “no”) were correctly applied, and there was no reversal issue in our evaluation. The decrease in scores can be rather attributed to the model's sensitivity [2] or interference among demonstrations [3] under in-context learning (ICL). In fact, it can even be noticed in the original GPT-3 paper [4] that additional demonstrations do not always lead to better performance and can indeed sometimes result in a notable decrease, demonstrating an inherent challenge in ICL. Taking the best, average, worst over different k-shot (0, 1, 5) configurations for each dataset to address the concern of sensitivity to some extent, we still observed significant gaps compared to the base one (please see the table of results below and Table 4 in Appendix).
>
> We updated our paper to include these clarifications to address your concerns. We firmly believe our results are correct. To further alleviate any doubts, we kindly refer you to Table 1 in Section 4.2 where we have extended our experiments to the financial domain, employing various (10) datasets to demonstrate the validity and versatility (up to 14.53 average gain).
>
> |               | Best               | Average            | Worst              |
> |---------------|--------------------|--------------------|--------------------|
> | **Dataset**       | **Base / Self-Spec.**  | **Base / Self-Spec.**  | **Base / Self-Spec.**  |
> | BioASQ-Factoid | 51.96 / **57.61**              | 43.47 / **50.00**                 | 30.9 / **37.35**                |
> | BioASQ-List    | **47.57** / 46.99              | 42.91 / **44.57**                 | 35.09 / **42.17**               |
> | BioASQ-Yesno   | 21.20 / **95.20**               | 13.60 / **91.49**                  | 8.80 / **85.27**                 |
> | PubMedQA           | **31.69** / 31.31              | 24.19 / **26.78**                 | 11.98 / **24.16**               |
> | AnatEM       | 9.63 / **21.25**               | 7.93 / **16.33**                  | 6.59 / **11.99**                |
> | BioNLP13CG         | 26.03 / **41.16**              | 24.19 / **32.63**                 | 21.76 / **24.93**               |
> | NCBI     | 27.88 / **46.54**              | 21.44 / **34.67**                 | **17.99** / 14.35               |
> | DDI            | 51.00 / **53.4**                  | 49.86 / **51.47**                 | 49.20 / **49.4**                 |
> | Medical Drugs  | 35.00 / **65.80**                  | 19.27 / **51.07**                 | 11.40 / **32.80**                |
> | HoC            | **62.84** / 62.65              | **26.40** / 25.42                  | 2.44 / **6.01**                 |
> | **Average**    | 36.48 / **52.19**              | 27.33 / **42.44**                 | 19.62 / **32.84**               |

---

> > ### Author Response · Authors · 2023-11-13
> >
> > >***The Rouge-L scores for the "DDI" and "Medical" datasets are identical to their respective F1 scores. I haven't investigated if this is possible, but please check the reported results carefully.***
> >
> > We appreciate your attention to the details in our reported results. Following your observation, we conducted a thorough re-examination of these specific cases and can confirm that the results are indeed accurate. The occurrence of identical values for both F1 and Rouge-L scores can arise in certain scenarios, particularly in classification tasks where the ground truths consist of single tokens. In such cases, both metrics, which fundamentally measure overlap between the predicted and true labels, can yield the same numerical value.
> >
> > >***During the seed generation phase, the proposed method requires the NLP benchmarks in the target domain. This limits the proposed approach to extend to other domains, as not all domains have this NLP benchmark accessible. With the paper using 80 seeds, one possible solution could be manually generating them; however, the paper didn't mention this point.***
> >
> > While we used established domain-specific datasets for seed construction to ensure their quality and relevance fairly, manual annotation is an assumed prerequisite for this initial step in self-alignment including existing works. However, we have demonstrated that effectiveness can be achieved with a minimal set of seeds (only 80), which we consider manageable and highly data-efficient. In fact, we have constructed manual seeds for sports domain (Appendix C) which we found not intensive. This requirement is notably less, by half, compared to prior works in general self-alignment [5, 6]. We further clarified this point in the paper as you suggested.
> >
> > >***How about the results if applying the proposed scheme to a larger model, such as LLaMA-65B? The base model used in the paper initially shows not good results (20-30 F1 scores), which are relatively easy to improve. If a larger model with better initial results is used, is it still feasible to have improvement through the proposed scheme?***
> >
> > We appreciate your question regarding the scalability of our method. Prior works on self-alignment leverage only larger models such as 175B and 65B, indicating that a larger scale of a base model is essential for the capability to create quality synthetic data. With this in mind, in our context, the investigation of a smaller scale model (e.g., 7B) was more prioritized as it was deemed a more challenging yet insightful endeavor due to its relatively limited knowledge/capability. As highlighted in Table 1 and Figure 7, our findings indicate the feasibility of self-specialization within such a scale, attesting to the method's practicality. We are optimistic that applying our self-specialization scheme to a larger model will yield improvements, albeit possibly with diminishing gaps by nature as the starting performance increases as pointed out. The improvement is anticipated especially considering that the self-specialized 30B already outperforms the 65B base model on the specialized domain by a large margin in Figure 4.
> >
> > >***Some places lack professional writing. For example, in Eq. (3), is introduced without prior definition. While the intended meaning may be inferred, a scientific paper should maintain consistency and rigor in its notation and explanations.***
> >
> > We acknowledge the importance of precision and clarity in writing. The oversight in Eq. (3) has been corrected, and we have ensured that all notations are consistently defined throughout the paper. We appreciate you bringing this to our attention.
> >
> > [1] Parmar, Mihir et al. “In-BoXBART: Get Instructions into Biomedical Multi-Task Learning.” NAACL 2022.
> >
> > [2] Zhao, Tony et al. “Calibrate Before Use: Improving Few-Shot Performance of Language Models.” ICML 2021.
> >
> > [3] Chen, Jiuhai et al. “How Many Demonstrations Do You Need for In-context Learning?” arXiv 2023.
> >
> > [4] Brown, Tom B. et al. “Language Models are Few-Shot Learners.” Neurips 2020.
> >
> > [5] Wang, Yizhong et al. “Self-Instruct: Aligning Language Models with Self-Generated Instructions.” ACL 2023.
> >
> > [6] Sun, Zhiqing et al. “Principle-Driven Self-Alignment of Language Models from Scratch with Minimal Human Supervision.” Neurips 2023.

---

> ### Author Response · Authors · 2023-11-21
> **Looking Forward to Your Reply**
>
> Dear Reviewer CV1V,
>
> Since the discussion period is ending tomorrow, we would greatly appreciate it if you could take a look at our response to your review and let us know if you have any remaining questions. Otherwise, we would really appreciate it if you could support the paper by increasing the score. We look forward to hearing from you and addressing any remaining concerns before the end of the discussion period.
>
> Best regards,
>
> Authors

---

> > ### Comment · Reviewer_CV1V · 2023-11-22
> >
> > Thanks for your response! I appreciate your effort to improve the paper.  However, this does not change the problem of the lack of novelty and technique contribution of the paper. I will my score unchanged.

---

> ### Author Response · Authors · 2023-11-22
> **Thank You for Your Reply!**
>
> Thank you for your reply! We are glad to see your recognition of our efforts to improve our paper based on your review. Unfortunately, we do not see any comments about the novelty or technical aspects in your review. Rather, it was primarily about the validity of experiments, leading to a score of 3. We sincerely thank you for your critical evaluation, as it has provided us with the opportunity to improve our paper. We believe our response along with additional experiments effectively addressed your concern mentioned in the review. We respectfully request a reconsideration of your assessment of our work. We remain open to any further feedback or clarifications you may deem necessary.

---

### Official Review · Reviewer_Kzpe · 2023-10-30

**Soundness:** 3 good
**Presentation:** 3 good
**Contribution:** 3 good
**Rating:** 6
**Confidence:** 3

**Summary:**

This paper focuses on self-alignment for expert domain specialization. It first conducts a benchmarking of existing aligned models, i.e., Alpaca-65B and Dromedary-65B, revealing that existing self-aligned models achieve marginal improvements compared to the original model. Then, it proposes a self-specialization method that leverages domain-specific unlabelled data and a few labeled seeds for the self-alignment process. The experiments on a medical domain show the proposed method outperforms its base model, and larger popular models as well.

**Strengths:**

1. The topic of specialization is important for deploying LLM to a specified domain.
2. The paper is well-written and easy to follow.
3. The idea of self-specialization is interesting, which utilizes both seed instructions and generated ones together.
4. Several experiments are conducted to evaluate the proposed method. The proposed self-specialization method outperforms the base model and larger LLaMA variants.

**Weaknesses:**

1. The benchmark only includes two aligned models. Better to include more aligned models for comprehensive benchmarking.
2. It claims that 'we hypothesize that the model expertise in different domains resides in “superposition” in the model’s parameters and hidden states'. But there are no further theoretical explanations or experimental results to support this claim. If the "superposition" could be explained in details, the application scope of the proposed method may be more clear.
3. The metrics of the y-axes should be added in figure 2 and 5.

**Questions:**

1. Why do you choose the two aligned models? Why not include other models for benchmarking?
2. Does the selection of seed instructions affect the final performance? If so, how do you eliminate this effect?

---

> ### Author Response · Authors · 2023-11-13
>
> We appreciate your positive reviews recognizing the importance of the topic, the quality of the paper, the idea, and the efforts on experiments. Down below, we address each of your comments and questions in detail.
>
> >***Why do you choose the two aligned models? Why not include other models for benchmarking?***
>
> Thank you for your inquiry regarding our choice of aligned models for benchmarking. Our decision was driven by specific criteria aligned with the key scenarios and objectives of our study. A central aspect of our research is the exploration of self-specialization under low-data scenarios. In this context, self-aligned models that are developed from scratch, such as Dromedary, are particularly relevant. These models represent a critical baseline for comparison as they do not rely extensively on large volumes of human-labeled data. At the time of our benchmarking, Dromedary was considered new and a strong model in this category. While Alpaca, which utilizes GPT-3.5 instead of LLaMA-65B for data generation, does not strictly fall under the category of self-aligned models, it was included to provide a benchmark for the upper bounds of performance achievable through automatic alignment methods.
>
> In response to Reviewer T5kV’s request, we have also conducted additional experiments comparing our self-specialized model with models that have undergone domain-specific pre-training (MedLLaMA-13B and PMC-LLaMA-7B/-13B, please see Figure 7 and Appendix C for the results). In short, the results indicate that our self-specialized 7B model is better than the 13B baselines including the one pre-trained on a huge domain-specific corpus (i.e., medicine) and the one further instruction-tuned using labeled data. Moreover, we also show (in the same Fig. 7, Appendix C) that our approach can be effectively applied on top of domain-specific pre-training demonstrating even further gains without requiring any additional (costly to collect) domain-specific data.
>
> >***It claims that 'we hypothesize that the model expertise in different domains resides in “superposition” in the model’s parameters and hidden states'. But there are no further theoretical explanations or experimental results to support this claim. ...***
>
> We appreciate your interest in our intuition. We concur that this concept, which posits that domain-specific knowledge exists in a latent form within LLMs, is a foundation of our approach. Our hypothesis is grounded in the idea that these models, due to their extensive pre-training on diverse datasets, inherently possess a broad spectrum of knowledge, which is not explicitly structured but rather intermingled within their parameters and hidden states.
>
> To empirically validate, we demonstrated the effectiveness and versatility of our self-specialization method across various domains (biomediacal and sports domains in the original paper, and also for the finance domain in response to Reviewer T5kV’s comment). These experiments show that by explicitly leveraging this latent knowledge, we can significantly enhance the model's performance in specific domains. For example, we successfully self-specialized the same LLaMA-2-7B base model to both biomed and finance domains enhancing its performance in both, thus supporting our hypotheses that these expertises have been “superimposed” inside that model, while our self-specialization approach helped surface them. These results serve as practical evidence supporting our hypothesis, illustrating that LLMs can be effectively aligned to specific domains despite the latent and unstructured nature of their knowledge.
>
> We believe that the theoretical exploration of this hypothesis would be both important and intriguing. However, such an investigation would extend beyond the scope of our initial study, which focuses on the practical application and empirical validation of the concept.
>
> >***The metrics of the y-axes should be added in figure 2 and 5.***
>
> Thank you for pointing this out! We explicitly added the used metrics, the F1 score, to figures 2 and 5.

---

> > ### Author Response · Authors · 2023-11-13
> >
> > >***Does the selection of seed instructions affect the final performance? If so, how do you eliminate this effect?***
> >
> > Thank you for raising a critical point regarding the influence of seed instruction selection on the final performance of our model. Intuitively, the choice of seed instructions plays a significant role in guiding the self-specialization process we believe, as they set the foundational context and direction for the model's learning trajectory. Fully exploring the end-to-end impact of seed instruction selection on model performance, however, is indeed resource-intensive. It involves extensive experimentation with various seed sets, which can be prohibitively expensive in terms of computational resources and time. Given these constraints, our approach has been to strike a balance between comprehensiveness and feasibility. While the choice of seed instructions does influence the model's learning trajectory, we have taken steps to ensure a balanced and representative selection: 1) Our seeds are curated to cover a broad spectrum of the target domain, minimizing bias towards any specific task. 2) We used established domain-specific datasets for seed selection, leveraging their quality and relevance. 3) We use a relatively small set of seeds, e.g. only 80 for biomedical, to minimize user burden and minimally constrain the model’s creativity.

---

> ### Author Response · Authors · 2023-11-21
> **Looking Forward to Your Reply**
>
> Dear Reviewer Kzpe,
>
> Since the discussion period is ending tomorrow, we would greatly appreciate it if you could take a look at our response to your review and let us know if you have any remaining questions. We look forward to hearing from you and addressing any remaining concerns before the end of the discussion period.
>
> Best regards,
>
> Authors

---

> > ### Comment · Reviewer_Kzpe · 2023-11-22
> > **Response to the comments from the authors**
> >
> > Dear authors,
> >
> > Thanks for the clarification. I also appreciate your efforts on improving the paper. I will keep my score and have no further questions.
> >
> > Best

---

### Official Review · Reviewer_T5kV · 2023-11-03

**Soundness:** 3 good
**Presentation:** 3 good
**Contribution:** 2 fair
**Rating:** 5
**Confidence:** 4

**Summary:**

This paper introduces a method called self-specialization, which extends the self-instruct method to a specific domain. The experimental results are evaluated on a set of biomedical tasks using two LLMs. The problem addressed in this paper is interesting and the experiment results show promise. However, the paper can be significantly improved by generalizing the proposed method to at least another domain, as the main contribution of the paper lies in the proposal of a method for domain specification. Additionally, the effectiveness of the proposed method, which is based on finetuning, needs to be clarified as the baseline performance is based on in-context learning. To improve the clarity of the method, an ablation study should be included, e.g., examining the contribution of domain-specific response generation to the overall performance.

**Strengths:**

- The paper is well-written and easy to follow, providing sufficient technical details. The authors also mention that the data, code, and trained model will be open source.
- The empirical findings demonstrate that incorporating unlabeled data positively impacts the model's ability to effectively respond to queries within a specialized domain, particularly in the challenging context of biomedical research.

**Weaknesses:**

- The proposed method, self-specialization, is an extension of the self-instruct [1] work to recover certain expertise in the LLMs. However, the method is only designed and evaluated for one specific domain without showcasing its generalization ability to other domains.
- The compared methods in the paper are based on zero-shot/few-shot settings, while the proposed method uses LoRA for finetuning. It would be beneficial to include stronger task-specific comparison methods to better illustrate the effectiveness of the proposed approach.
- Since the method has multiple components, it would be helpful to show the contribution of each component to the overall performance. For example, how does domain-specific response generation impact the final results?
- The performance in the knowledge sparse domain is uncertain. It is understood that domain response generation involving harnessing knowledge is one of the main contributors to the specification, but it is important to understand the method's bottleneck when existing knowledge is limited for certain domains.
- The paper should discuss potential data contamination and address how the authors ensure that the data for downstream testing does not overlap with the generated data.

[1] Wang, Yizhong, et al. "Self-instruct: Aligning language model with self generated instructions." arXiv preprint arXiv:2212.10560 (2022).

**Questions:**

See above.

---

> ### Author Response · Authors · 2023-11-13
>
> Thank you very much for your thoughtful and constructive review of our work. We appreciate your positive comments recognizing the quality of our paper and our findings, as well as your concerns to improve our work. Down below, we address each of your comments in detail.
>
> >***The method is only designed and evaluated for one specific domain without showcasing its generalization ability to other domains.***
>
> We understand your concern about the generalizability of our method to other domains. To address your concern, we have evaluated our Self-Specialization method on the finance domain (10 popular financial NLP datasets covering 6 tasks), demonstrating similar effectiveness (up to 14.53 average gain) to the one observed for biomedicine. The finance domain results have been added to Table 1. This new result, along with our original findings in biomedicine, showcases the versatility of our self-specialization approach across varied specialization domains.
>
> >***The compared methods in the paper are based on zero-shot/few-shot settings, while the proposed method uses LoRA for finetuning. …***
>
> In our study, the primary objective was to enhance the base model's domain-specific capabilities through self-specialization, a process inherently different from conventional fine-tuning approaches. Although the process utilizes LoRA for specialization, it is important to note that our approach fundamentally relies on synthetic data generated by the model itself. This unique aspect sets our method apart, as it effectively starts from scratch, focusing on self-generated, domain-specific instructional data for low-data scenarios. Finally, the base model and the base model improved through our Self-Specialization (using synthetic self-generated data) are compared fairly in the same zero-shot/few-shot setting. Hence we believe the comparison is fair.
>
> Nevertheless, in response to your valuable feedback, we have conducted additional experiments comparing our self-specialized model with models that have undergone domain-specific pre-training (MedLLaMA-13B and PMC-LLaMA-7B/-13B, please see Figure 7 and Appendix C for the results). In short, the results indicate that our self-specialized 7B model is better than the 13B baselines including the one pre-trained on a huge domain-specific corpus (i.e., medicine) and the one further instruction-tuned using labeled data. In the same Figure (Fig. 7, Appendix C) we also show that our Self-Specialization can be effectively applied on top of the domain specific pre-training demonstrating further improvements, thus underlining even more its practical potential.
>
> > ***Since the method has multiple components, it would be helpful to show the contribution of each component to the overall performance. …***
>
> For the component contribution, we kindly refer the reviewer to section 4.3 where we provide ablations & analyses including the effect of domain-specific response generation (i.e., external knowledge).

---

> > ### Author Response · Authors · 2023-11-13
> >
> > >***The performance in the knowledge sparse domain is uncertain. …***
> >
> > Thank you for providing invaluable feedback. We indeed acknowledge the challenge in a domain where the base LLM may lack substantial domain knowledge. This challenge, in fact, was the key motivation for our work, which reveals the marginal effect of general self-alignment on a specific domain, and thus harnesses the base model to align it to a targeted domain using domain-specific seeds and additional external knowledge.
> >
> > To further address your concern, we have explored the application of our method to a smaller size base model (LLaMA-2-7B), which inherently possesses less domain knowledge (Figure 7 and Appendix C). We also extended these experiments beyond the biomedical domain, specifically also testing in the financial domain (Table 1). Moreover, we showed qualitative analyses in the sports domain. In all experiments, Self-Specialization demonstrates significant gains (up to 14.53% on average) even for the “less-domain-knoweldgeable” LLaMA-2-7B. These experiments demonstrate the versatility and potential of our method across various domains, including those with limited existing knowledge in the base model.
> >
> > >***The paper should discuss potential data contamination and address how the authors ensure that the data for downstream testing does not overlap with the generated data.***
> >
> > Thank you for raising the general issue of potential data contamination. We appreciate the opportunity to clarify this aspect of our research. In our study, the synthetic data generated by the base model is indeed intended to mirror the distribution of the targeted domain ideally. This approach is integral to the primary goal of our self-specialization method, which aims to align the base model with the specific nuances and knowledge of the target domain. We can confidently assert that the test data has not been explicitly encountered by the model during the self-specialization process. That is, the test samples were NOT used for seed examples or for anything else in the self-specialization process.
> >
> > Whether the base model had seen similar (to test) data or not during pre-training is we believe non-consequential, as the comparison is between models (base and base+self-specialization) that have a shared pre-training condition and hence is fair. Finally, in response to the reviewer’s comment, we have manually verified a large sample of the generated data and manually compared to a sample of the test data and have not observed any overlaps. This is reasonable given the high randomness of the generation process involving creative brainstorming and sampling for decoding.

---

> ### Author Response · Authors · 2023-11-21
> **Looking Forward to Your Reply**
>
> Dear Reviewer T5kV,
>
> Since the discussion period is ending tomorrow, we would greatly appreciate it if you could take a look at our response to your review and let us know if you have any remaining questions. Otherwise, we would really appreciate it if you could support the paper by increasing the score. We look forward to hearing from you and addressing any remaining concerns before the end of the discussion period.
>
> Best regards,
>
> Authors

---

> > ### Comment · Reviewer_T5kV · 2023-11-22
> >
> > Thanks for your detailed response! I will keep my score since the method's performance in evaluated domains, and contributions from different modules are not convincing, as well as model's overall performance can be significantly attributed to the data overlap.

---

> ### Author Response · Authors · 2023-11-23
> **Thank You for Your Reply**
>
> We thank the reviewer for their thoughtful comment, yet we would like to respectfully disagree. The whole point of our self-specialization approach is to improve the model’s specialized domain performance using self-generated synthetic data. The fine-tuning data in our approach is self-generated by the model and hence there can be no overlap between fine-tuning data and target domain data. If the reviewer is worried about any potential overlap between target domain test data and the base model pre-training, then please note that we always compare the base model performance and our self-specialized model performance. Intuitively, both models share the same real data pre-training by design, hence our improvement (using only synthetic self-generated data) is completely fair. This is further strengthened by the fact that our main specialization performance gains could be obtained without any retrieval - on synthetic self-generated data alone. No matter whether the reviewer agrees with our arguments, we would like to thank the reviewer for giving us the opportunity to improve our paper.

---

### Author Response · Authors · 2023-11-13
**Paper Update with Additional Experiments**

Dear Reviewers,

We appreciate all of your valuable comments on our work. We are pleased to share the updated version of our paper with additional experiments on the following:
- **Table 1 in Section 4.2** - Experiments on the **finance domain** beyond biomedicine, showing significant gains (up to 14.53 average gain) on 10 popular financial NLP datasets.
- **Figure 7 in Appendix C** - demonstrating Self-Specialization with an **additional model variant/size** (LLaMA-2-7B) and **additional comparison with existing models** including the one pre-trained on a huge domain-specific corpus (MedLLaMA-13B) and the one further instruction-tuned after the pre-training (PMC-LLaMA-7B/-13B). In Appendix C, we also demonstrate how Self-Specialization can be applied on top of the domain-specific pre-training consistently adding further gains (without requiring any additional and costly domain-specific data collection).

We hope this will address the concerns you may have regarding **generalization, validity of experiments, model scales, baselines, and so on**.
We respond in detail to each of your individual comments below.

---

### Author Response · Authors · 2023-11-18
**Gentle Reminder**

Dear Reviewers,

In response to your valuable feedback, we have conducted significant additional experiments and updated our paper and rebuttal accordingly. However, we were somewhat discouraged to see the lack of engagement from the reviewers. We are keen to understand your perspectives on these updates and would welcome any further comments or questions you might have to facilitate the discussion.